# The Significance of Cathepsin B in Mediating Radiation Resistance in Colon Carcinoma Cell Line (Caco-2)

**DOI:** 10.3390/ijms242216146

**Published:** 2023-11-09

**Authors:** Ramadan F. Abdelaziz, Ahmed M. Hussein, Mohamed H. Kotob, Christina Weiss, Krzysztof Chelminski, Christian R. Studenik, Mohammed Aufy

**Affiliations:** 1Division of Pharmacology and Toxicology, Department of Pharmaceutical Sciences, University of Vienna, 1090 Vienna, Austria; a01576752@unet.univie.ac.at (R.F.A.); ahmed.hussein@univie.ac.at (A.M.H.); kotobm84@univie.ac.at (M.H.K.); a01546450@unet.univie.ac.at (C.W.); mohammed.aufy@univie.ac.at (M.A.); 2Division of Human Health, International Atomic Energy Agency, Wagramer Str. 5, 1400 Vienna, Austria; k.chelminski@iaea.org

**Keywords:** colon cancer, cathepsin B, radiotherapy, radioresistance, cathepsin inhibitors, CA074

## Abstract

Cathepsins (Caths) are lysosomal proteases that participate in various physiological and pathological processes. Accumulating evidence suggests that caths play a multifaceted role in cancer progression and radiotherapy resistance responses. Their proteolytic activity influences the tumor’s response to radiation by affecting oxygenation, nutrient availability, and immune cell infiltration within the tumor microenvironment. Cathepsin-mediated DNA repair mechanisms can promote radioresistance in cancer cells, limiting the efficacy of radiotherapy. Additionally, caths have been associated with the activation of prosurvival signaling pathways, such as PI3K/Akt and NF-κB, which can confer resistance to radiation-induced cell death. However, the effectiveness of radiotherapy can be limited by intrinsic or acquired resistance mechanisms in cancer cells. In this study, the regulation and expression of cathepsin B (cath B) in the colon carcinoma cell line (caco-2) before and after exposure to radiation were investigated. Cells were exposed to escalating ionizing radiation doses (2 Gy, 4 Gy, 6 Gy, 8 Gy, and 10 Gy). Analysis of protein expression, in vitro labeling using activity-based probes DCG04, and cath B pull-down revealed a radiation-induced up-regulation of cathepsin B in a dose-independent manner. Proteolytic inhibition of cathepsin B by cathepsin B specific inhibitor CA074 has increased the cytotoxic effect and cell death due to ionizing irradiation treatment in caco-2 cells. Similar results were also obtained after cathepsin B knockout by CRISPR CAS9. Furthermore, upon exposure to radiation treatment, the inhibition of cath B led to a significant upregulation in the expression of the proapoptotic protein BAX, while it induced a significant reduction in the expression of the antiapoptotic protein BCL-2. These results showed that cathepsin B could contribute to ionizing radiation resistance, and the abolishment of cathepsin B, either by inhibition of its proteolytic activity or expression, has increased the caco-2 cells susceptibility to ionizing irradiation.

## 1. Introduction

Within several enzymes housed in lysosomes, caths stand out as a remarkable family of lysosomal proteases, displaying an expansive range of functions. caths play a vital role in intracellular housekeeping, including their involvement in antigen processing during immune responses. Additionally, they contribute to the degradation of various proteins and chemokines, thereby actively involved in maintaining cellular homeostasis (as reviewed in [1,2,3,4]).

Caths encompass several types, including serine types (caths A and G), aspartic acid types (caths D and E), and the most abundant cysteine types (caths B, C, F, H, K, L, O, S, V, W, and X) [3,5,6]. These diverse caths play crucial roles in numerous cellular activities, such as hormone synthesis and activation, as well as some physiological processes like apoptosis and autophagy.

Regarding cancer development, secreted caths play a role in the degradation and remodeling of the tumor extracellular matrix (ECM), while intracellular caths serve as crucial components of signaling pathways that can promote cell growth and inflammation [6,7]. Furthermore, caths are involved in response to anticancer therapy within the tumor microenvironment and can play pivotal roles in the development of resistance to therapeutic interventions [8,9,10,11,12].

Moreover, a study has identified caths within the tumor microenvironment as potential contributors to resistance during radiotherapy [13].

Given that the primary mechanism of action of radiotherapy is the selective damage to cancer cell DNA, evaluating the involvement of caths in modulating radiation-induced DNA damage and repair mechanisms is of great significance in predicting the efficacy of radiotherapy. The expression and activity of caths can be influenced by various factors, including radiation-induced stress responses and the tumor microenvironment.

Caths have demonstrated their ability to regulate tumor cell survival and apoptosis following radiation exposure. Cath-mediated proteolysis can activate or deactivate specific apoptotic pathways, thus influencing the fate of irradiated cells [14]. Studies have shown that knockdown of cath L using external inhibitors or siRNA-mediated silencing resulted in increased radiosensitivity of the p53-mutant glioma cell line U251. This effect could be attributed to G2/M phase cell cycle arrest or the malfunction of DNA damage repair processes [15].

Moreover, previous findings suggest that radiation induces cathepsin S expression through ROS-IFN-gamma pathways, and this heightened expression may play a role in the development of radioresistance [16], it was also investigated that cath H signaling plays a crucial role in regulating the cancer metabolic switch and apoptosis, contributing to the development of radioresistance in Hepatocellular Carcinoma cells (HCC). These findings indicate the potential significance of cath H in the diagnosis and therapeutic approaches for HCC [17].

Similarly to our study, a previous study showed that cath B plays a pivotal role in promoting radioresistance by elevating homologous recombination activity in glioblastoma (U87 and U251 cells). The introduction of cath B knockdown results in increased radiosensitivity, attributed to a reduction in homologous recombination (HR) efficiency. Further investigations unveiled that the knockdown of cath B induces cell cycle arrest in G0/G1 phases, consequently impairing HR efficiency [18].

Cath B serves as a cysteine proteinase and exists in various forms, including a single-chain variant with a molecular mass of approximately 31 kDa, a heavy-chain form featuring a complete C-terminal end weighing 25 kDa and a heavy-chain variant with a C-terminal end at 23.4 kDa [19,20]. Cath B typically operates within acidic lysosomes to facilitate protein degradation. However, in various human diseases cath B undergoes translocation to the cytosol, characterized by a neutral pH. In this cytosolic environment, the enzyme becomes activated, triggering inflammation and cell death [21].

Nonetheless, cath B’s control can undergo modifications at various stages, leading to its increased expression and extracellular release. This phenomenon potentially indicates cath B’s involvement in changes that contribute to the advancement of cancer [22,23,24,25,26].

Cath B’s influence extends to intracellular communications, autophagy induction, and immune resistance. Its involvement in cell survival varies significantly across different conditions, ranging from caspase-dependent apoptosis to aiding tumor neovascularization and metastasis. Furthermore, the potential involvement of cath B in non-neoplastic diseases was discussed. In addition to the development of cath B specific inhibitors such as small molecular inhibitors and the exploration of microRNA targeting cath B, with the aim of assessing their viability for clinical applications [19].

In this research, our aim was to investigate the role of cath B in influencing the development of radioresistance in colon carcinoma cells (caco-2). Furthermore, we explored the impact of a targeted cath B inhibitor and the utilization of CRISPR/Cas9 technology to inhibit/knockout cath B on enhancing radiosensitivity. Therefore, targeting cath B may represent a promising approach to sensitize Caco-2 cells to radiation and improve the overall therapeutic outcomes in future in vivo or clinical studies. Further investigations are needed to fully elucidate the molecular interactions and regulatory networks involving cath B with the aim of developing targeted interventions that can enhance the effectiveness of radiotherapy in the clinical setting.

## 2. Results

### 2.1. Active-Site Labelling of Cysteine Peptidases with DCG-04 in Cultured Caco-2

Caco-2 cells were subjected to increasing doses of ionizing irradiation followed by treatment with DCG-04, an activity-based probe designed for papain family cysteine caths. DCG-04 contains an epoxide warhead that selectively and irreversibly interacts with the active site cysteine residue of targeted cysteine caths, facilitating the study of their specific activities rather than their abundance. After cell harvesting, SDS-PAGE under reducing conditions was performed to visualize DCG-04 labelled cysteine peptidases using streptavidin-HRP. Four major bands with molecular weights of 70, 40, 30, and 25 KDa were observed (Figure 1).

Considering that lysosomal caths typically ranging in molecular weights between 20 and 45 KDa, the highest 70 KDa band was disregarded. The other three bands co-migrated with different forms of cath B, namely pro-cath B, intermediate form, and mature form. This co-migration pattern strongly indicates that cath B, a key member of the papain family cysteine caths, may be closely involved in the cellular response to ionizing irradiation.

### 2.2. Pulldown of Cath B

To investigate the potential upregulation of cath B in response to ionizing irradiation, caco-2 cell lysates were obtained after exposure to different ionizing irradiation doses. Ex vivo labelled, with DCG-04 before radiation exposure was, followed by incubation with streptavidin agarose beads. The proteins bound to the streptavidin beads were eluted and subjected to SDS-PAGE and immunoblotting using specific antibodies targeting cath B. The results showed that cath B showed significant upregulation across all tested doses (*p* < 0.001). At 2 Gy, Cath B exhibited approximately a four-fold upregulation, while at 8 Gy, it was upregulated by over seven-fold. At the highest tested dose of 10 Gy, Cath B showed a nearly nine-fold upregulation (Figure 2).

Notably, although the upregulation generally increased with higher doses, it was not strictly dose-dependent, as the upregulation at 2 Gy was slightly higher than that at 4 Gy.

### 2.3. Expression of Cath B before and after Exposure to Ionizing Irradiation

A series of experiments were conducted to detect the expression of cath B before and after exposure to various ionizing irradiation doses. Following the exposure, cells were harvested and protein extraction was performed. Subsequently, protein electrophoresis and immunoblotting were carried out using specific antibodies targeting cath B, while β-tubulin served as the loading control (Figure 3).

The obtained results demonstrated that the expressions of cath B at 2 Gy, 4 Gy, 6 Gy, and 8 Gy were more than three times greater than those in non-irradiated cells. Interestingly, at 2 Gy, cath B expression was slightly higher compared to 4 Gy and 6 Gy. However, the most significant upregulation of cath B was observed at 10 Gy, where it exceeded 4.5 times the expression in non-irradiated cells.

Notably, the upregulation of cath B was observed in both its pro-cath B and processed/mature forms. However, in contrast to the total cath B levels, the mature cath B was less abundant at the 10 Gy irradiation compared to the smaller doses.

### 2.4. Implication of Cath B Inhibition on Ionizing Radiation Cytotoxicity

To investigate the potential involvement of cath B in ionizing radiation cytotoxicity, caco-2 cells were exposed to ascending doses of ionizing radiation in the presence and absence of the cath B specific inhibitor CA074.

The MTT assay results demonstrated an escalation in cytotoxicity attributed to elevated ionizing radiation doses. Interestingly, pre-treating Caco-2 cells with a cath B inhibitor further augmented the cytotoxic impact of radiation and correspondingly reduced cell viability. For instance, at 2 Gy, the effect of ionizing radiation on cell viability was relatively mild (around 5%), but it was increased to approximately 18% after cath B inhibition. While at 4 Gy, cell viability was reduced to about 93%, which was further decreased to about 77% after blocking the proteolytic activity of cath B with the specific inhibitor (Figure 4).

Similarly, at 6 Gy, the cytotoxic effect of ionizing radiation was more pronounced, resulting in a cell viability of approximately 64% without cath B inhibitor and 55% after cath B inhibition. At 8 Gy, cell viability was inhibited to 58% in the absence of the cath B inhibitor and 49% in its presence. Finally, at 10 Gy, cell viability was reduced to 40% without cath B inhibition and 36% after cath B inhibition. However, no significant differences were observed between the two control groups.

### 2.5. Impact of Cath B CRIPSR/CAS9 Mediated Knockout on Ionizing Radiation Cytotoxicity

To validate the previous cytotoxicity findings, we created caco-2 cells with cath B knockout using CRISPR/Cas9 technology. The expression of cath B was examined in both the wild-type caco-2 cells and the caco-2 cells with cath B knockout. In the wild-type caco-2 cells, cath B was detected, whereas in the caco-2 cells with cath B knockout, cath B was not detectable (Figure 5).

Notably, knocking out cath B alone had no significant effect on cell viability. However, when caco-2 cells were exposed to 2 Gy of radiation, the cell viability decreased to approximately 80% compared to the controls. On the other hand, when cath B was knocked down, the cell viability was reduced to about 20%. Similarly, with 4 Gy radiation dose, cell viability decreased to about 66%, while knocking down cath B resulted in a viability of about 20%. At 6 Gy radiation dose, cell viability dropped to about 56%, but cath B knockout further reduced cell viability to approximately 18%. Remarkably, at 8 Gy and 10 Gy radiation doses, there were similar outcomes, with cell viability reaching about 45%, while cath B knockout in caco-2 cells led to a significant reduction in cell viability to about 10% (Figure 6).

### 2.6. Impact of Cath B Inhibition on Apoptosis in Ionizing Radiation Exposed Cells

In order to investigate the potential impact of cath B inhibition on the expression of apoptotic proteins, caco-2 cells were exposed to increasing doses of ionizing radiation in the presence and absence of the cath B inhibitor CA074. After exposure, the cells were harvested and proteins were extracted for analysis. Protein electrophoresis was performed under reducing conditions, followed by immunoblotting on nitrocellulose membranes using specific antibodies targeting the pro-apoptotic protein BAX and the anti-apoptotic protein BCL-2 (Figure 7).

The expression of BCL-2 was significantly reduced in CA074-treated cells compared to cells exposed only to ionizing radiation, except for the 10 Gy dose where the effect was less pronounced. At 4 Gy, BCL-2 expression decreased by approximately 40% in CA074-treated cells compared to those exposed only to ionizing radiation. The reduction was more pronounced at 6 Gy and 8 Gy doses, reaching about 50% in CA074-treated cells compared to cells exposed only to ionizing radiation.

Regarding the pro-apoptotic protein BAX, its expression increased by about 25% at 2 Gy and 6 Gy doses. However, the effect was less pronounced at 4 Gy and 8 Gy doses, although the difference was still statistically significant. Remarkably, at 10 Gy, BAX expression was slightly reduced in cells treated with cath B inhibitor in comparison with cells exposed only to ionizing radiation.

These findings provide evidence of a potential role of cath B in caco-2 cells’ resistance to ionizing radiation treatment and suggest a possible mechanism involving the suppression of certain apoptotic events.

## 3. Discussion

Colon cancer primarily impacts the elderly population, yet it is not limited to a specific age group. Its onset commonly originates from tiny cell clusters known as polyps that develop within the colon. Although most polyps are non-cancerous, certain ones may transform into colon cancer as time progresses. Typically, polyps remain asymptomatic [27]. Colon cancer is regarded as one of the most perilous and life-threatening forms of cancer.

Radiation therapy is employed as a treatment method to effectively manage colon cancer. Several single-institution retrospective studies have demonstrated enhanced local control and potentially improved survival rates by incorporating external irradiation and/or intraoperative radiation in the treatment approach [28].

Caths constitute a wide-ranging group of peptidases primarily active within endosomes and lysosomes, crucial components of the lysosomal death pathway [29]. Central to this pathway is lysosomal membrane permeation (LMP), wherein the integrity of the lysosomal membrane is compromised. This results in the release of luminal contents, including caths, into the cytoplasm. The unleashed caths then initiate a sequence of events leading to organelle impairment and triggering various forms of cell demise—encompassing apoptosis, pyroptosis, or necroptosis [30,31].

Recent investigations have also underlined the connection between LMP and radiosensitivity. The induction of lysosomal biogenesis by irradiation amplifies the discharge of lysosomal hydrolysates into the cytoplasm, consequently intensifying cell mortality and achieving radiosensitization [32]. Importantly, the influence of caths on cell demise hinges on whether their cleavage of substrates is activating or deactivating in nature.

Previous studies have observed an increase in lysosomal caths as a consequence of radiotherapy, which has been associated with a crucial role in promoting resistance to this form of treatment [13,16,18]. Specifically, in glioblastoma cancer, upregulation of cath B has been linked to radiotherapy resistance [13,18]. This was in agreement with our results when the expression of all protein bands shown in Figure 2 significantly increased upon exposure to any of the tested irradiation doses, indicating upregulation of many cysteine peptidases in response to ionizing irradiation. This finding suggests that ionizing irradiation may play a role in altering cellular processes related to these peptidases, potentially impacting cellular functions. Interestingly, our own investigation discovered a dose-independent increase in cath B expression in caco-2 cells after ionizing radiation exposure, with some doses leading to a tenfold rise in expression (Figure 3). These data support the previous results that many cysteine peptidases are upregulated as a consequence of exposure to ionizing irradiation. These findings suggest a complex regulatory response of cath B to ionizing irradiation.

By investigating the specific activities and abundance of cystine cathepsins with active site labeling we found that the expression of all bands significantly increased upon exposure to any of the tested irradiation doses, indicating upregulation of many cysteine peptidases in response to ionizing irradiation. This finding suggests that ionizing irradiation may play a role in altering cellular processes related to these peptidases, potentially impacting cellular functions. Further research is warranted to explore the specific mechanisms through which ionizing irradiation influences the regulation of cysteine peptidases and its implications for cellular behavior.

Considering that lysosomal caths typically range in molecular weights between 20 and 45 KDa, the highest 70 KDa band was disregarded. The other three bands co-migrated with different forms of cath B, namely pro-Cath B, intermediate form, and mature form. This co-migration pattern strongly indicates that cath B, a key member of the papain family of cysteine caths, may be closely involved in the cellular response to ionizing irradiation [11].

In order to explore the potential increase in cath B levels in reaction to ionizing irradiation, lysates from caco-2 cells were collected subsequent to exposure to varying doses of ionizing radiation. It was observed that cath B demonstrated substantial upregulation across all doses tested. These data support the previous results that many cysteine peptidases are upregulated as a consequence of exposure to ionizing irradiation [13] and suggest a complex regulatory response of cath B to ionizing irradiation. Further investigations are needed to understand the underlying mechanisms responsible for this non-linear upregulation pattern.

To verify whether the increase in activity is attributed to higher expression levels of cath B, the increase in cath B expression was detected in both pro-Cath B and processed/mature forms. Yet, in contradiction to the overall cath B levels, the mature cath B exhibited lower abundance at the 10 Gy irradiation dose compared to the smaller doses. Our findings suggest a correlation between increased cath B activity and its expression, indicating that exposing caco-2 cells to irradiation leads to elevated cath B expression, indicating that the upregulation of cath B was not strictly dose-dependent and these results agree with a previous study [33].

In addition, to validate the pivotal role of cath B in irradiation-induced cytotoxicity and its contribution to the resistance of caco-2 cells against ionizing radiation, we utilized CRISPR/CAS9 to completely abolish cath B expression or employed the cath B-specific inhibitor CA07. The sensitivity of the caco-2 cells to ionizing radiation significantly increased. Notably, the impact was more pronounced in caco-2 cells with cath B knocked down compared to cells with cath B inhibited. This could be attributed to CRISPR/CAS9 fully eliminating cath B expression, whereas CA074 mostly but not fully inhibits its activity within the cells. Consistent with our research, a prior study highlighted the crucial involvement of cath B in enhancing radioresistance. This effect was attributed to the elevation of homologous recombination activity specifically observed in glioblastoma cells. Moreover, and in agreement with our findings, the process of knocking down cath B, resulting in a reduction of its expression, significantly contributed to increasing radiosensitivity in the experimental setting [18]. To assess the potential influence of cath B inhibition on the expression of apoptotic proteins, caco-2 cells were subjected to escalating doses of ionizing radiation in the presence and absence of the cath B inhibitor CA074. These findings provide evidence of a potential role of cath B in caco-2 cells’ resistance to ionizing radiation treatment and suggest a possible mechanism involving the suppression of certain apoptotic events.

Moreover, in the cath B inhibited caco-2 cells, the pro-apoptotic protein BAX showed a significant increase, while the expression of the anti-apoptotic protein BCL-2 decreased, as mentioned in our results. This suggests that apoptosis is more likely to occur when cells are exposed to weak or moderate radiation doses, while necrosis is triggered at higher radiation doses [34].

It is important to note that cath B is not the sole factor contributing to cell resistance to radiotherapy, as presented in our study. Other investigations have highlighted the role of cath S, for instance, as an important player in promoting resistance to radiotherapy in certain tumors [16]. Furthermore, the inhibition of Cathepsin L has been found to enhance the radiosensitivity of human glioma U251 cells. This heightened radiosensitivity is associated with the induction of G2/M cell cycle arrest and an increase in DNA damage. These findings underscore the significant role of Cathepsin L in influencing cellular responses to radiation in glioma cells [15], as well as the reversal of radioresistance in hepatocellular carcinoma achieved through the knockdown of Cathepsin H, marking a significant breakthrough in therapeutic approaches. This reversal is intricately linked to a metabolic switch within the cellular environment, ultimately culminating in the activation of apoptotic pathways [17].

In conclusion, these data support the previous results that many of cysteine peptidases, in particular cath B, are upregulated as a consequence of exposure to ionizing irradiation. These findings suggest a complex regulatory response of cath B to ionizing radiation, and suppression of cath B before exposure to ionizing radiation treatment, not only increases radiation sensitivity but also activated pathways leading to cell death.

Further investigations are needed to understand the underlying mechanisms responsible for this non-linear upregulation pattern. Targeting lysosomal caths shows promise as a potential strategy for enhancing the efficacy of radiotherapy and overcoming radiotherapy resistance in cancer treatment. However, further research is warranted to explore the specific mechanisms through which ionizing irradiation influence the regulation of cysteine peptidases and its implications on cellular behaviour.

## 4. Materials and Methods

### 4.1. Caco-2 Cell Culture

Caco-2 cells derived from human colon carcinoma were used in this study (Given by Dr. Rosa Lemmens-Gruber, Vienna University). These cells were cultured in Dulbecco’s Eagle’s medium (DMEM, Gibco, ThermoFisher Scientific, Waltham, MA, USA), containing 4 mM glutamine. Additionally, 100 units/mL of penicillin and 100 μg/mL of streptomycin (ThermoFisher Scientific, Waltham, MA, USA) were supplemented as antibiotics, and the cells were incubated at 37 °C in an environment with 5% CO_2_.

### 4.2. Radiation Treatment

Two groups of caco-2 cells treated and untreated with cath B specific inhibitor CA074 (Merck, Rockville, MD, USA) were irradiated using a 6 MV photon beam of a medical linear accelerator (LINAC) (Varian Medical Systems, Palo Alto, CA, USA) with different doses (2, 4, 6, 8 and 10 Gy, with constant dose rate of 3 Gy/min). For assessment of delivered dose the Eclipse (Varian) treatment planning system (TPS) was used. Another two groups of Caco-2 cells treated and untreated with cath B inhibitor were not exposed to radiation and served as negative control. Caco-2 cell lines (~107) were treated for 24 h at 37 °C in a complete medium containing 10 μM CA074 before radiation. Dimethylsulphoxide solvent (final concentration 0.1%) was added to the untreated groups.

### 4.3. MTT Assay, Cell Viability Cytotoxicity Test

Cells were placed into 96-well plates, with 2000 cells per well along with 100 μL of medium, and then incubated for a duration of 24 h. Subsequently, the cells were treated with inhibitors (CA074) individually, using concentrations of 10 μM. The viability of the cells after 24 h of treatment was assessed using a 3-(4,5-dimethylthiazol-2-yl)-2 and -5 diphenyltetrazolium bromide (MTT)-based viability assay called EZ4U (Biomedica, Vienna, Austria). For this assay, 20 μL of EZ4U solution was introduced into each well, followed by a 2 h incubation at 37 °C. The absorbance was then measured at 450 nm using a microplate reader (Infinite F200, Tecan, Männedorf, Switzerland), with 620 nm serving as a reference for unspecific background values. This entire experimental procedure was replicated three times, with each repetition consisting of triplicate samples [35,36,37].

### 4.4. Western Blotting

The experiments were carried out following the procedures outlined in prior references [38,39,40,41,42]. Initially, cells were cultured after radiation in 100 mM cell culture dishes within a 5% CO_2_ incubator, using DMEM medium supplemented with 5% fetal bovine serum (ThermoFisher Scientific, Waltham, MA, USA). Afterward, the medium was aspirated, and the cells were washed twice with PBS. Following this, cells were gently detached using lysis buffer (comprising 200 mM sodium acetate, 150 mM NaCL, pH 5.5, and supplemented with 40 μM E-64), and subsequently transferred to 1.5 μL centrifuge tubes.

For homogenization, the cells were treated with ultrasonication (10 s then 30 s on ice/3 times) while maintained on ice, followed by the addition of 0.1% Triton X-100. The homogenized cell mixture was then incubated on ice for 30 min. Subsequently, the samples were cleared by centrifugation at 15,000× *g* for 10 min. The separated proteins were subjected to 12.5% SDS-PAGE under reducing conditions. These proteins were subsequently transferred onto nitrocellulose membranes (obtained from Santa Cruz Biotechnology, Dallas, TX, USA) using semi-dry blotting at 25 V for 30 min.

To prevent non-specific binding, the membrane was treated with a blocking solution (3% BSA in PBS) for duration of 3 h. Following this, the membrane was exposed to primary antibodies, namely cath B (ThermoFisher Scientific, Waltham, MA, USA) (1:2000), BAX and Bcl-2 (Cell Signaling, Graz, Austria) (1:1000), and β-tubulin (Sigma Aldrich, St. Louis, MO, USA) (1:3000), for a period of 90 min. Subsequently, the membrane was washed five times with PBST and then incubated for an additional 90 min with the corresponding secondary antibodies. This was followed by three washes with PBST and one wash with PBS.

For visualization, enhanced chemiluminescence (Amersham ECL plus Western blotting detection reagent, GE Healthcare, Vienna, Austria) was employed. The membranes were exposed to X-ray films (Amersham Hyper film ECL, GE Healthcare, Vienna, Austria), and the resulting experimental films were scanned and quantified using ImageJ software, 1.54f (NIH, Bethesda, MD, USA).

### 4.5. Active Site Labelling of Cysteine Caths

In order to label cysteine caths within cultured cells, the utilization of the activity-based probe DCG04 (Medkoo, Morrisville, NC, USA) has proven effective. This probe is essentially a biotinylated variant of the general cysteine peptidase inhibitor E-64, selectively designed for targeting cysteine peptidases. Notably, DCG04 has the unique capacity to selectively bind to active cysteine peptidases present within complex protein mixtures [43].

To label the cells after radiation treatment, caco-2 cells were incubated for 72 h at 37 °C with 10 μM DCG04. Following this incubation, cellular protein extract was meticulously prepared. Subsequently, 30 μg of this extract underwent separation through 12.5% SDS polyacrylamide gels. The separated proteins were then transferred onto a nitrocellulose membrane sourced from Santa Cruz Biotechnology (Dallas, TX, USA). To ensure specificity and minimize non-specific binding, the transferred proteins were subjected to a blocking step using a solution of 3% bovine serum albumin (BSA) from ThermoFisher Scientific (Waltham, MA, USA) in PBS. The membrane was then exposed to streptavidin-horseradish peroxidase (0.125 μg/mL in PBST) from BioLegend (San Diego, CA, USA). This crucial step preceded the application of enhanced chemiluminescence detection methods, allowing for the visualization of the labeled cysteine caths.

### 4.6. Pull-Down of DCG04 Labelled Cysteine Caths

For the purpose of isolating in vitro labelled cysteine caths, the following procedure was executed. Initially, 250 μL (equivalent to approximately 400 μg) of cellular protein extracts were quantified utilizing the Bradford method as outlined by Bradford in 1976 [44,45].

Subsequently, the cellular extracts previously labelled with DCG04 were appropriately diluted by combining them with 750 μL of a binding buffer comprising 20 mM sodium acetate (pH 5.5), 150 mM sodium chloride, 0.1% triton X-100, 10 μg/mL E-64, 10 μg/mL leupeptin sourced from Sigma Aldrich (St. Louis, MO, USA), and 1 mM PMSF from Abcam. This mixture was subjected to centrifugation at 14,000× *g* for duration of five min, with the resulting supernatant incubated overnight along with 40 μL of settled streptavidin beads at 4 °C.

To facilitate further processing, the beads were collected via centrifugation lasting 5 min at 3000× *g*. Subsequently, a sequence of washes was carried out: five washes employing 20 mM sodium acetate (pH 5.5), 150 mM sodium chloride, and 0.1% triton X-100, followed by two additional washes using 10 mM Tris-HCL at pH 6.8 [34].

The settled beads were then mixed with 40 μL of a 2× sample buffer and heated for a period of five min at 95 °C [35,36]. The ensuing supernatant underwent separation through SDS-PAGE, followed by blotting onto a nitrocellulose membrane. This membrane was subsequently subjected to immunoblotting utilizing antibodies (1:2000 dilution) obtained from ThermoFisher Scientific (Waltham, MA, USA) and specifically directed against cath B.

### 4.7. Cathepsin B Knockout Experiment

Twenty-four h post-seeding and cell confluency reached 70–90%, caco-2 cells underwent transfection with either the CRISPR/Cas9 plasmid targeting cath B (Santa Cruz sc-400360) or the corresponding control plasmid (Santa Cruz sc-418922). The transfection procedure was conducted using the X-tremeGENE ™ HP DNA transfection reagent sourced from Roche Diagnostics (catalog Nr. 6366244001, Mannheim, Germany) and strictly adhering to the guidelines stipulated by the manufacturer as follows: X-tremeGENE™ HP DNA transfection reagent, plasmid DNA and diluent were left to warm with gentile vortex. The diluent and plasmid DNA were added and gently mixed in a sterile tube and then X-tremeGENE™ HP DNA transfection reagent was added later to the diluted DNA. The final Mixture was incubated for 15 min at 25 °C and then was added to Caco-2 cells in a dropwise manner with Gentile shaking for proper distribution. The cells were incubated with the mixture cells for 72 h before the exposure to different doses of ionizing radiation [42,46].

### 4.8. Statistical Analysis

A statistical analysis was conducted employing a non-parametric *t*-test to compare two groups. For scenarios involving more than two groups, either a one-way ANOVA or a two-way ANOVA was employed based on the number of independent variables. While a significant result from an analysis of variance (ANOVA) F-test provides overall evidence of group differences, it does not specify which pairs of means exhibit divergence. To discern specific differences among three or more group means, post hoc tests were employed. Following the ANOVA, Dunnett’s multiple comparison test was utilized to pinpoint pairs with significant differences. This approach compares means from multiple experimental groups against a single control group mean to identify any disparities.

Furthermore, to mitigate the risk of false positives, the Bonferroni test was implemented. This adjustment, pioneered by Bonferroni, serves to maintain the integrity of statistical significance assessments. All statistical analyses were executed using GraphPad Prism version 6.00 for windows, by GraphPad Software2, San Diego, CA, USA, in conjunction with Microsoft Excel 365. Significance was established at a probability level of *p* < 0.05. Data were presented as mean ± standard error (SE). For detailed statistical parameters relevant to specific experiments, please refer to the appropriate sections or figure legends.

## Figures and Tables

**Figure 1 ijms-24-16146-f001:**
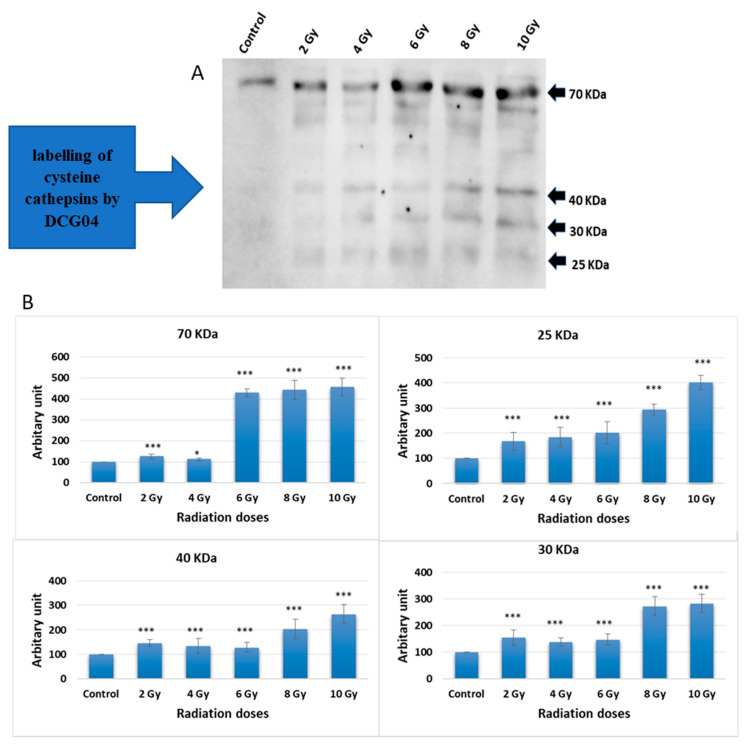
Radiation treatment influence on the active site labelling of cysteine caths by DCG04. (**A**) Active-site labelling with DCG04. Protein samples were subjected to protein electrophoresis and Western blotting with streptavidin–horseradish peroxidase. (**B**) Cellular content of 25 KDa and 30 KDa, 40 KDa and 70 KDa bands in caco-2 irradiated and DCG-04 labelled cells were compared to DCG-04 only labelled cells (Control). The data were analysed using two-way ANOVA with Dunnett’s post hoc analysis (* *p* < 0.05; *** *p* < 0.001; N = 5). Statistics were calculated using GraphPad Prism.

**Figure 2 ijms-24-16146-f002:**
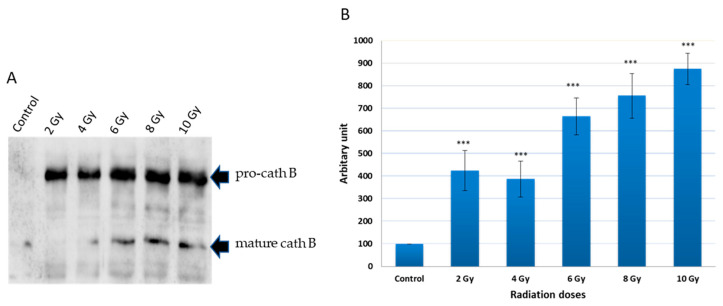
Avidin pull-down experiment of cysteine caths with and without radiation treatment. (**A**) Avidin pull-down experiment was performed as described in materials and methods. Conjugated proteins to avidin Sepharose beads were subjected to SDS-PAGE and Western blotting with antibodies specific to human cath B. (**B**) Ratio of cellular contents of cath B with and without radiation treatment. The data were analyzed by *t*-test and statistics using GraphPad Prism. *** *p* < 0.001. Graphs are shown as mean ± SE, N = 5.

**Figure 3 ijms-24-16146-f003:**
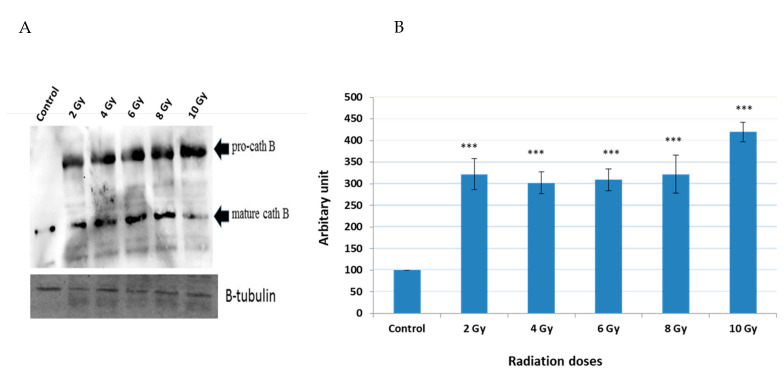
Cath B expression evaluation with and without different doses of radiation. (**A**) Western blot analysis of Cath B expression. (**B**) Ratio of cellular contents of cath B with and without radiation treatment. The data were analyzed by *t*-test and statistics using GraphPad Prism. *** *p* < 0.001. Graphs are shown as mean ± SE, N = 5.1.

**Figure 4 ijms-24-16146-f004:**
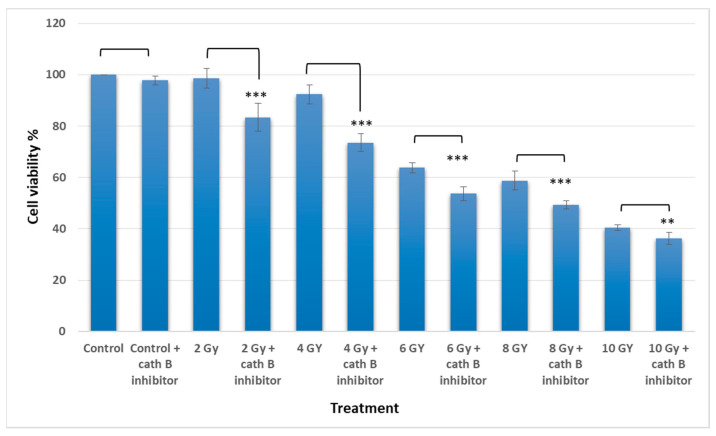
Cytotoxicity analysis for treated and no treated cells with radiation and cath B inhibitor. The data were analyzed by *t*-test and statistics using GraphPad Prism. ** *p* < 0.01; *** *p* < 0.001. Graphs are shown as mean ± SE, N = 8.

**Figure 5 ijms-24-16146-f005:**
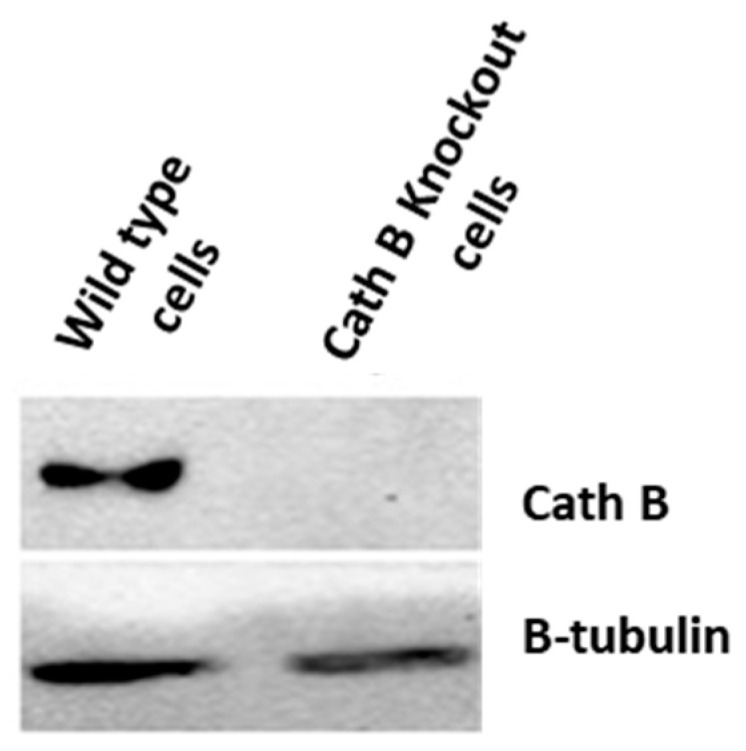
Cathepsin B knockout impact on non-irradiated and irradiated Caco-2 cells.

**Figure 6 ijms-24-16146-f006:**
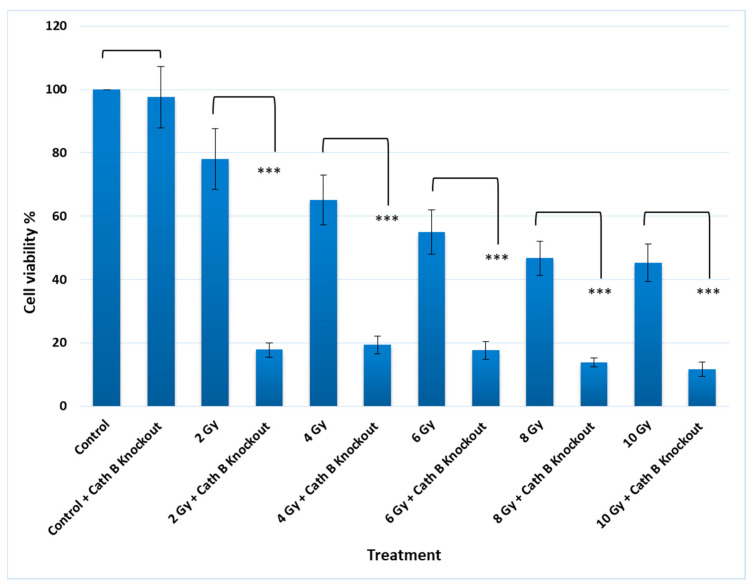
Cathepsin B knockout impact on non-irradiated and irradiated cells toxicity. The data were analysed by *t*-test and statistics using GraphPad Prism. *** *p* < 0.001. Graphs are shown as mean ± SE, N = 8.

**Figure 7 ijms-24-16146-f007:**
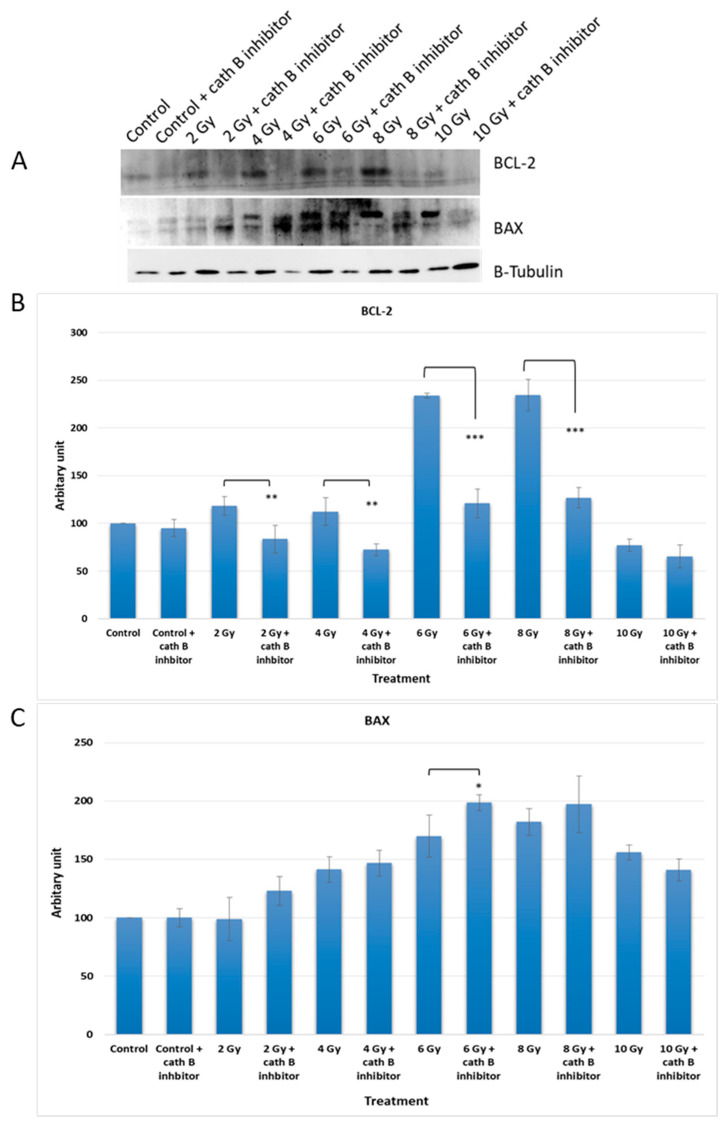
Cath B inhibition impact on apoptosis rate in treated and non-treated cells with ionizing radiation. (**A**) Conducted Western blot analysis for BCL-2 and BAX. (**B**) Examined the cellular content ratio of BCL-2. (**C**) Investigated the cellular content ratio of BAX. The data were analysed using two-way ANOVA with Dunnett’s post hoc analysis (* *p* < 0.05; ** *p* < 0.01; *** *p* < 0.001; N = 3). Statistics were calculated using GraphPad Prism.

## Data Availability

Data is contained within the article.

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
