# Peer review of "The Significance of Cathepsin B in Mediating Radiation Resistance in Colon Carcinoma Cell Line (Caco-2)"

_ijms, 2023, doi:10.3390/ijms242216146_

Round 1

Reviewer 1 Report

Comments and Suggestions for Authors

The authors Ramadan F. Abdelaziz, Ahmed M. Hussein, Mohamed H. Kotob, Christina Weiss, Krzysztof Chelminski, Christian R. Studenik and Mohammed Aufy have submitted  a manuscript (ID: ijms-2682562) entitled “The Significance of Cathepsin B in Mediating Radiation Resistance in Colon Carcinoma cell line (Caco-2) to the IJMS section: Molecular Pathology, Diagnostics, and Therapeutics. I have found this article as a non-peer reviewed draft in the  internet including an instructive graphical abstract which I have missed in the actual manuscript. The published results are sound and will reach the general readership of the target section. Since solid structural data of cathepsin B are available the structure – function relationship of this class of molecules in relation to the reported radiation resistance could  have been discussed in more detail. Furthermore, a recent article about this topic (Wang et. al. (2023) The Role of Cathepsin B in Pathophysiologies of Non-tumor and Tumor tissues: A  Systematic Review”. Journal of Cancer 14(12): 2344-2358. doi: 10.7150/jca.86531) is also not disussed.

Author Response

Dear colleagues at MDPI, IJMS Editorial Office, Dear respective reviewers

Thank you for affording us the chance to submit a revised version of our manuscript, "[The Significance of Cathepsin B in Mediating Radiation Resistance in Colon Carcinoma Cell Line (Caco-2)]." We extend our gratitude for the time and dedication you and the reviewers invested in offering constructive feedback. The insightful comments provided by the reviewers have been invaluable, and we have diligently incorporated changes to address most of their suggestions. These modifications are clearly delineated within the manuscript. Below is a detailed response addressing each of the reviewers' comments and concerns.

Note: The revised manuscript was done in tracking changes style, therefore, lines and references numbers will be change once you accept the changes.

Comments from Reviewer 1

Dear reviewer,

Thank you for taking the time to review our manuscript and share your valuable suggestions. We have diligently addressed all of your concerns in the revised manuscript file. Below, you'll find detailed responses to each of your comments.

Comment 1: Since solid structural data of cathepsin B are available the structure – function relationship of this class of molecules in relation to the reported radiation resistance could have been discussed in more detail.

Response: Well noted, agreed and addressed within the manuscript (please see lines, 114-134, 148- 152, 394-399).

Comment 2: Furthermore, a recent article about this topic (Wang et. al. (2023) “The Role of Cathepsin B in Pathophysiologies of Non-tumor and Tumor tissues: A Systematic Review”. Journal of Cancer 14(12): 2344-2358. doi: 10.7150/jca.86531) is also not discussed.

Response: Thank you for sharing this great article, which has very valuable data regarding cathepsin B and its different roles, we could incorporate some relevant information within the text and cited as a reference. (Please see lines 148- 152 and was cited as reference 25).

Reviewer 2 Report

Comments and Suggestions for Authors

My comments

Title: The Significance of Cathepsin B in Mediating Radiation Resistance in Colon Carcinoma Cell Line (Caco-2)

In this manuscript, the author investigated the significance of Cathepsin B in mediating radiation resistance in colon carcinoma cell lines. They reported that cathepsin B could contribute to ionizing radiation resistance and abolishing of cathepsin B either by inhibiting its proteolytic activity or expression has increased in caco-2 cells susceptibility to ionizing irradiation. I need authors to reply to these comments to get final acceptance for their valuable data.

  1. The abstract must be clear, and the information is irrelevant to this study.   
  2. The introduction must be clear and clear to the study's aim. The less is more. So, the author should discuss the originality of the manuscript.
  3. Experiments are very limited. Why did the author not test the apoptosis DNA damage or Mitochondrial study for further confirmations? These studies are not enough to conclude the activity.
  4. The author should add how the dose of Cathepsin B inhibitor was fixed in this study.
  5. Why has the author not tested in vivo models? 
Comments on the Quality of English Language

 Minor editing of the English language is required.

Author Response

Dear colleagues at MDPI, IJMS Editorial Office, Dear respective reviewers

Thank you for affording us the chance to submit a revised version of our manuscript, "[The Significance of Cathepsin B in Mediating Radiation Resistance in Colon Carcinoma Cell Line (Caco-2)]." We extend our gratitude for the time and dedication you and the reviewers invested in offering constructive feedback. The insightful comments provided by the reviewers have been invaluable, and we have diligently incorporated changes to address most of their suggestions. These modifications are clearly delineated within the manuscript. Below is a detailed response addressing each of the reviewers' comments and concerns.

Note: The revised manuscript was done in tracking changes style, therefore, lines and references numbers will be change once you accept the changes.

Dear reviewer,

Thank you for dedicating your time and effort to reviewing our manuscript and providing valuable suggestions. Your insights are instrumental in enhancing the clarity and quality of our work. To address your concerns, we have made comprehensive revisions, which are reflected in the updated manuscript file. Please see the below response to your comments.

Comment 1: The abstract must be clear, and the information is irrelevant to this study. 

Response:  Thank you so much for the great observation, agreed and addressed accordingly with more relevant results.

Comment 2: The introduction must be clear and clear to the study's aim. The less is more. So, the author should discuss the originality of the manuscript.

Response: Thank you and well noted, agreed and by this observation we could update the introduction accordingly, please see lines  (114-134, 148- 152, 148- 152). Much appreciated.

Comment 3: Experiments are very limited. Why did the author not test the apoptosis DNA damage or Mitochondrial study for further confirmations? These studies are not enough to conclude the activity.

Response: Thank you for the constructive observation. However, we believe the theory of the study is confirmed by different methodologies, and these specific methodologies for DNA damage can be considered for further studies. In the same regard, we would like to note the relevant information from our study and others as below:

  • (A) The DNA damage and apoptosis are regular response from cells to ionizing radiation:

Through ionization, chemical reactions are triggered, causing significant disturbances to various cell molecules, particularly DNA (Szatkowska and Krupa 2020; Oyefeso et al. 2023; Burgio, Piscitelli, and Migliore 2018).

The DNA damage induced by radiation sets off a signalling transduction pathway known as the DNA damage response (DDR). This activation involves multiple cellular signalling molecules determining the cell's fate, which may include cell cycle arrest, apoptosis, senescence, autophagy, and DNA repair (Santivasi and Xia 2014; Vignard, Mirey, and Salles 2013; Murmann-Konda et al. 2021).

References:

  • Szatkowska, M., and R. Krupa. 2020. 'Regulation of DNA Damage Response and Homologous Recombination Repair by microRNA in Human Cells Exposed to Ionizing Radiation', Cancers (Basel), 12.
  • Burgio, E., P. Piscitelli, and L. Migliore. 'Ionizing Radiation and Human Health: Reviewing Models of Exposure and Mechanisms of Cellular Damage. An Epigenetic Perspective', Int J Environ Res Public Health, 15.
  • Oyefeso, F. A., G. Goldberg, Nyps Opoku, M. Vazquez, A. Bertucci, Z. Chen, C. Wang, A. R. Muotri, and M. J. Pecaut. 'Effects of acute low-moderate dose ionizing radiation to human brain organoids', PLoS One, 18: e0282958.
  • Santivasi, W. L., and F. Xia. 2014. 'Ionizing radiation-induced DNA damage, response, and repair', Antioxid Redox Signal, 21: 251-9.
  • Vignard, Julien, Gladys Mirey, and Bernard Salles. 'Ionizing-radiation induced DNA double-strand breaks: a direct and indirect lighting up', Radiotherapy and Oncology, 108: 362-69.
  • Murmann-Konda, T., A. Soni, M. Stuschke, and G. Iliakis. 'Analysis of chromatid-break-repair detects a homologous recombination to non-homologous end-joining switch with increasing load of DNA double-strand breaks', Mutat Res Genet Toxicol Environ Mutagen, 867: 503372.

  • (B) Mitochondrial studies:

  • In this study we performed MTT Assay, Cell Cytotoxicity Test. And EZ4U kits was used in this regard, the technique relies on the observation that viable cells have the capacity to transform mildly colored or colorless tetrazolium salts into highly colored formazan derivatives. This reduction mechanism is reliant on the presence of functional mitochondria. Consequently, the EZ4U assay is closely associated with the mitochondrial pathway of apoptosis.
  • In our study, we placed our primary focus on investigating the expression of the proapoptotic protein BAX, as depicted in Figure 7. BAX plays a pivotal role as a regulator in the intrinsic pathway of apoptosis. When cells are exposed to apoptotic signals, BAX becomes activated and forms oligomers on the mitochondrial outer membrane (MOM), ultimately leading to the permeabilization of the MOM. This MOM permeabilization represents a critical event in the process of apoptosis. It's important to note that an increase in BAX expression can trigger cell death through the initiation of the mitochondrial permeability transition. Consequently, the upregulation of BAX serves as an indicator of the involvement of the mitochondrial pathway in apoptosis.
  • In addition to BAX, we also assessed the expression of the antiapoptotic protein BCL-2, as shown in Figure 7. BCL-2 is a member of the apoptosis regulators, but it serves other functions as well. This protein interacts with partners that act as inhibitors of cell death, and collectively, they regulate and mediate the intricate process by which mitochondria contribute to cell death, known as the intrinsic apoptosis pathway.

Comment 4: The author should add how the dose of Cathepsin B inhibitor was fixed in this study.

Response: thank you for this observation, the concentration of 10 uM CA-074 is commonly reported in the literature as an effective dose to inhibit cathepsin B activity in various experiments, suggesting that it is often used to achieve a substantial inhibition of cathepsin B activity in a given context. Please see below relevant references:

  • Mirković B, Markelc B, Butinar M, Mitrović A, Sosič I, Gobec S, Vasiljeva O, Turk B, Čemažar M, Serša G, Kos J. Nitroxoline impairs tumor progression in vitro and in vivo by regulating cathepsin B activity. Oncotarget. 2015 Aug 7;6(22):19027-42. doi: 10.18632/oncotarget.3699. PMID: 25848918; PMCID: PMC4662473.
  • Yoon MC, Christy MP, Phan VV, Gerwick WH, Hook G, O'Donoghue AJ, Hook V. Molecular Features of CA-074 pH-Dependent Inhibition of Cathepsin B. Biochemistry. 2022 Feb 15;61(4):228-238. doi: 10.1021/acs.biochem.1c00684. Epub 2022 Feb 4. PMID: 35119840; PMCID: PMC9096814.
  • Matarrese P, Ascione B, Ciarlo L, Vona R, Leonetti C, Scarsella M, Mileo AM, Catricalà C, Paggi MG, Malorni W. Cathepsin B inhibition interferes with metastatic potential of human melanoma: an in vitro and in vivo study. Mol Cancer. 2010 Aug 4;9:207. doi: 10.1186/1476-4598-9-207. PMID: 20684763; PMCID: PMC2925371.
  • Szpaderska AM, Frankfater A. An intracellular form of cathepsin B contributes to invasiveness in cancer. Cancer Res. 2001 Apr 15;61(8):3493-500. PMID: 11309313.
  • Mihalik R, Imre G, Petak I, Szende B, Kopper L. Cathepsin B-independent abrogation of cell death by CA-074-OMe upstream of lysosomal breakdown. Cell Death Differ. 2004 Dec;11(12):1357-60. doi: 10.1038/sj.cdd.4401493. PMID: 15297886.

Comment 5: Why has the author not tested in vivo models?

Response 5:  Thank you for the great suggestion, which is promising idea, however we believe that can be for future study and to expand our research area and planning for upcoming projects.

Round 2

Reviewer 2 Report

Comments and Suggestions for Authors

The authors have satisfactorily responded to all comments and made the necessary changes to the manuscript.